# Enhanced Readout System for Timepix3-Based Detectors in Large-Scale Scientific Facilities

**DOI:** 10.3390/s25061860

**Published:** 2025-03-17

**Authors:** Petr Burian, Benedikt Bergmann, Pavel Broulím, Martin Farkaš, Tomáš Kulhánek, Petr Mánek, Ondřej Růžička, Petr Smolyanskiy, Ondřej Urban, Jan Zich

**Affiliations:** 1Faculty of Electrical Engineering, University of West Bohemia, Univerzitni 26, 301 00 Pilsen, Czech Republic; broulimp@fel.zcu.cz (P.B.); farkasm@fel.zcu.cz (M.F.); tkulhane@fel.zcu.cz (T.K.); ruzickao@fel.zcu.cz (O.R.); urbano@fel.zcu.cz (O.U.); zichj@fel.zcu.cz (J.Z.); 2Institute of Experimental and Applied Physics, Czech Technical University in Prague, Husova 240/5, 110 00 Prague, Czech Republic; benedikt.bergmann@utef.cvut.cz (B.B.); petr.manek@utef.cvut.cz (P.M.); petr.smolyanskiy@cvut.cz (P.S.)

**Keywords:** Timepix3, readout system, detectors

## Abstract

This work presents a novel readout system for the Timepix3-based detector, designed specifically for large-scale facilities, such as particle accelerators. The authors begin by outlining the challenges associated with utilizing the Timepix3 detector over long distances. This paper proposes a solution to these challenges and describes its practical implementation. Beam test results are provided to demonstrate the system’s performance, with a particular focus on time-of-flight measurements. Additionally, the authors address the complexities of operating in challenging environments, particularly those involving radiation and magnetic fields.

## 1. Introduction

The Timepix3 ASIC readout chip [1] has already demonstrated excellent results and benefits across numerous projects, including precise measurement of half-lives, particle tracking, and time-of-flight measurements [2,3,4,5]. It has been employed as a radiation monitor in the ATLAS Experiment at CERN [6], and its performance in various low-power modes has also been well documented [7]. However, the integration of a larger number of (well-synchronized) Timepix3 detectors in environments with large research infrastructures—typically accelerators—remains challenging. To address this, we have developed a novel readout system tailored for applications requiring high data rates, long distances, and operation in harsh radiation fields. While other similar systems exist [8,9], they are designed for specific experimental conditions and rely on specialized components developed at CERN. The proposed system is intended to be more versatile, allowing the use of either commercial or CERN-based components. Additionally, the authors aimed to develop a device fully compatible with their existing ecosystem, including chipboards and control software [10] developed by their research groups. Experience has shown that proper software support is crucial for the successful adoption of a device among users.

In this paper, the authors first discuss issues with using the detector over long distances and in harsh environments (radiation and magnetic fields). Then, possible approaches to face these challenges are suggested. The authors present their concept and implementation of the designed readout system, together with results obtained during the test beam campaign.

## 2. Timepix3 Readout ASIC

### 2.1. Main Features

Timepix3 is the second generation of readout ASIC for pixel detectors in the Timepix family. It was developed within the Medipix3 Collaboration [11] at CERN as the successor to the Timepix detector [12]. The detector comprises two fundamental components: the readout ASIC chip and the active sensor layer, interconnected via bump bonding (see Figure 1). The active sensor layer can be fabricated from various materials, such as silicon (Si), cadmium telluride (CdTe), and gallium arsenide (GaAs). The ASIC chip employs 130 nm CMOS technology and integrates 65,536 channels, forming a 256 × 256 pixel matrix with a pixel pitch of 55 µm. Unlike the Timepix detector, Timepix3 features simultaneous measurement of Time over Threshold (ToT), the energy, and the Time of Arrival (ToA) of detected particles. The time binning provided by the chip is 1.5625 ns, using a 40 MHz clock generator (with a period of 25 ns) for coarse ToA measurement, combined with a 640 MHz clock from a ring oscillator for fine ToA (fToA) measurement. Accurate time-stamping necessitates time-walk correction, which can be accomplished experimentally as described in [13], or through test pulses, as demonstrated in [14].

The Timepix3 detector offers two readout modes: frame-based with zero suppression and data-driven. In frame-based mode, the entire pixel matrix of the sensor is read out, excluding pixels with zero values. The maximum achievable frame rate in this mode is approximately 1300 frames per second (fps). In data-driven mode, individual events (hit pixels) are transmitted to the output bus upon triggering. The primary advantage of this scheme is that most of the pixel matrix remains active, with only the triggered pixel becoming inactive for approximately 475 ns. Subsequently, the pixel is ready to register another hit. This configuration allows for a hit rate of up to 40 million hits per second, per square centimeter (Mhits/s/cm^2^).

### 2.2. Long-Distance Use

As suggested in the previous section, Timepix3 is a promising tool for radiation monitoring, providing precise energy measurements and timestamping for individual particles. However, questions arise regarding its suitability for applications where the detector and readout system must be spatially separated. This separation is crucial in scenarios where the readout system needs to be protected from potential damage caused by radiation or strong magnetic fields. These challenges, particularly those related to radiation, are inherent to the environments of scientific infrastructures such as accelerators.

In [6], we presented a concept where the Timepix3 detector is interconnected with the readout system via very long cabling. In this setup, only the detector, mounted on a printed circuit board (PCB) with voltage regulators, is placed within the radiation field, while the readout system remains in a safe area, shielded from damage. However, increasing the distance and cable length introduces limitations to the maximum achievable data rate between the detector and the readout system due to signal attenuation in the cables. Although the detector’s output utilizes SLVS signals with 8B/10B encoding—a significant improvement over older Timepix detectors (and even the newer Timepix2)—the maximum data rate is still constrained by distance. At distances of approximately 100 m, we achieved a maximum data rate of 40 Mbps, corresponding to 625 kHits/s per line.

The detector’s input signals, which are necessary for its control and for providing the reference clock, are implemented as pure SLVS/LVDS signals without DC-balanced encoding. However, since these input signals operate at only 40 MHz, they are less sensitive to attenuation compared to the output signals operating at 640 MHz.

To fully exploit the speed capabilities of Timepix3, the current concept must be modified. Two potential solutions are available: incorporating signal repeaters into the detector’s data output lines or translating the output signals to a different physical layer where the length of the transmission medium no longer imposes a speed limitation.

## 3. Proposed Concept

In an ideal scenario, a chipboard with a Timepix3 assembly could be equipped with a fiber transceiver, translating the detector’s serial output data to an optical signal and vice versa (optical data to the detector’s serial input). This setup would allow for an almost unlimited distance between the chipboard and the readout system (although power supply via cable would still be required). Unfortunately, due to specific features of Timepix3, this idea is not straightforward from an implementation perspective.

The primary challenge lies in the fact that Timepix3’s output is implemented through eight SLVS data lanes, each operating at a maximum rate of 640 Mbps. This data rate is not typical for direct optical transceiver input. Additionally, directly transmitting these signals via fiber would require using all eight data links, which, while feasible, significantly complicates chipboard design. Alternatively, output data could be accumulated and merged into a single data stream, requiring only one high-speed output link. However, this elegant solution necessitates a sophisticated device capable of aggregating Timepix3 data from all output lines into a unified stream. Such functionality, likely implemented using an FPGA, introduces additional complexity and challenges. Using an FPGA on the chipboard would require extra voltage regulators, memory components (for FPGA configuration), and other elements, increasing sensitivity to radiation and magnetic fields. This undermines the original idea of limiting components exposed to the radiation field to only essential elements.

Our proposed concept combines the advantages of the aforementioned approaches while mitigating their drawbacks. The readout system is divided into two parts: the front-end (Data Concentrator—DC) and the back-end (BE), as illustrated in Figure 2.

The back-end component (BE) serves as the interface between the computer and the Data Concentrator (DC). It also handles various functionalities, including detector control, low-level data processing, and synchronization with other elements of the measurement chain. The BE is expected to be in an area with low radiation, where fully commercial designs and components can be utilized. This allows for the integration of advanced, powerful devices capable of performing complex data processing directly in hardware.

## 4. Implementation

This section outlines the current implementation of the proposed concept. However, each project and measurement chain has its own specific requirements. For this reason, the concept should be viewed as a flexible, generic platform that can be adapted to meet the needs of individual projects.

### 4.1. Detector Unit (Chipboard)

For particle tracking, we designed two-layer Timepix3 detector units (chipboards; see Figure 3) tailored for use in environments with harsh radiation. A pair of Timepix3 detectors is arranged face to face, with their sensors oriented towards each other. This configuration allows the insertion of foils between them, which can serve as neutron converters or separators. Measurement with this unit is described in detail in [15].

From an electronics perspective, only essential components are used: the Timepix3 chips and power supplies. Voltage regulation is implemented by LHC4913 linear regulators, known for their ability to withstand high radiation doses over their operational lifespan [16].

For connectivity, six RJ-45 connectors are employed due to their high reliability and wide availability. Each Timepix3 chip utilizes four output data signals, enabling a maximum hit rate of 40 MHit/s per layer, resulting in 80 MHit/s per unit. To minimize the number of required signals, most control signals (except the shutter signal) are shared between the two detectors. The selection of a specific Timepix3 chip is based on its unique chip ID.

While the massive electrical shielding of RJ-45 connectors may present challenges in environments with extremely high radiation levels, experience with these connectors in previous ATLAS experiment setups (published in [6,17]) has demonstrated their reliability under the expected radiation conditions of this system.

The detector unit is evidently the most radiation-exposed component of the system, with an expected annual dose of up to 3 kGy. Two primary effects caused by radiation are anticipated.

First, single-event upsets (SEUs) may occur in the pixel matrix configuration. To mitigate this, periodic reconfiguration of all configuration data is planned, a method successfully implemented in a previous project [6]. SEUs can lead to blinking or noisy pixels, generating excessive data rates that could eventually saturate the communication bandwidth. While periodic reconfiguration should help mitigate this issue, it will not eliminate SEUs. The second anticipated effect is an increase in the sensor’s leakage current, indicating degradation. Additionally, changes in the gain of the input amplifiers may occur, necessitating a repeat of the threshold equalization process and recalibration over time—for example, during the winter shutdown.

Based on our experience, and despite some expected degradation, the Timepix3 ASIC, sensor, and associated chipboard electronics can withstand the radiation levels corresponding to the annual dose.

### 4.2. Data Concentrator (Front-End Device)

The key component of the designed DC (see Figure 4) is the Microsemi MPF300TS FPGA, which manages all tasks related to data manipulation between the detectors and the BE. Thanks to its flash-based technology for storing the bitstream, this FPGA offers enhanced radiation hardness. Testing has demonstrated that the 28 nm SONOS-based architecture remains functional up to 300 krad (SiO_2_), with negligible degradation. Additionally, single-event latch-up (SEL) testing revealed a high threshold, while single-event upset (SEU) rates for SRAM and flip-flops were relatively low [18].

The DC can communicate with three detector units as described above, accommodating a total of six Timepix3 detectors. The primary connection to the BE is established via a pair of SFP+ slots designed for 10GbE transceiver modules, currently using MM-OM3 (Multi-Mode) or SM (Single-Mode) fiber. This setup provides a total data rate of 20 Gbps for data transfer.

In addition to data handling, the DC includes several auxiliary functions. It features an independent voltage regulator to supply power to the detector units. A The high voltage for detector bias can either be generated using add-on modules (providing three independent bias voltages) or supplied by an external source. The add-on modules can generate up to 500 V for both polarities and support leakage current measurement. The selection of the bias source and enabling the power supply for detectors are fully controlled by commands issued by the BE.

For synchronization, clock distribution, and diagnostics, the DC is equipped with a pair of RJ-45 connectors transmitting LVDS signals.

It should be noted that the switching voltage regulators currently used in the DC are not highly resistant to magnetic fields. The authors propose replacing these commercial regulators, which are well proven in physics instrumentation, with specialized power supplies using air-core coils (e.g., bPOL48V designed in CERN [19,20]). Similarly, radiation-hardened transceivers and fibers, such as the CERN-developed Versatile Link+ system [21], should replace commercial SFP+ modules when the DC is deployed in a harsh radiation environment.

Our initial motivation was to utilize the DC in the toroid area (subjected to a magnetic field of up to 4T) of the ATLAS experiment, located relatively far from the interaction point (IP). According to estimations, the annual radiation dose should not exceed 10 Gy. Therefore, we tested several models of commercial SFP+ modules under a comparable total ionizing dose (TID) of 50 Gy, using a proton beam flux of 2.2 × 10^8^ cm^−2^s^−1^. As expected, single-event upsets (SEUs) are anticipated.

### 4.3. Back-End Device

Because the BE does not require specific interfaces or functionalities (from an HW viewpoint; all operations are performed using an FPGA design), any development kit featuring an FPGA device, SFP+, GPIO port, and a suitable communication interface can be utilized. This flexibility accelerates the development process and results in financial savings. For their implementation, the authors used the Terasic HAN Pilot development kit [22], which is based on an Intel ARRIA10 FPGA device. Gigabit Ethernet is employed for system control, while PCI Express Gen3 4x ensures the fast transfer of measured data.

### 4.4. Data Rate Limits

Assuming the utilization of three detector units producing a maximal hit rate of 80 MHits/s each, the total data transfer requirement reaches 240 MHits/s, corresponding to a data rate of 1.44 GB/s (11.52 Gbps). This demand is fully met by the pair of 10 Gbps links interconnecting the DC and the BE. However, the actual hit rate will be higher due to the need to transfer additional Time-of-Arrival (ToA) packets. For transferring measured data to the computer, the PCIe interface provides a real data rate of up to 3.3 GB/s. Nevertheless, due to driver limitations, the authors achieved a maximum of 1.9 GB/s under the Windows operating system. For applications where high data rates are not required, users can rely on Gigabit Ethernet for both system control and data transfer.

### 4.5. Synchronization

Exact time synchronization among detectors is a mandatory and key feature of the designed system. Although the reference clocks produced by the DC are identical for all detectors in the system, differences in cabling and clock phases generated by the PLLs in Timepix3 can cause time offsets in the nodes of the measurement chain. Unfortunately, Timepix3 does not have a direct trigger input. To address this issue, the DC monitors the resynchronized shutter register in the Timepix3 ASIC via the PLL outputs of the Timepix3 chips. These outputs provide the current value of the shutter register outside chip. Based on these signals, the DC then generates a specialized signal consisting of pulses, whose relative positions indicate the time offsets between detectors. This signal is transmitted to the BE via metallic synchronization cables, where it can be measured using either the internal TDC implemented in the FPGA or an external TDC device. In other words, the exact start times of the detectors are recorded, allowing for off-line compensation of timing delays.

This approach is particularly advantageous when multiple DCs need to be used and synchronized. It is important to note that a precise static time analysis of key components (entities) in the FPGA design is essential to successfully compensate (off-line) for all time offsets and delays.

The synchronization scheme is shown in Figure 5. The diagram and the following text explain the synchronization concept of the system with six detector layers (numbered 0 to 5), which is discussed in detail in Section 5.2.

Key timing-related commands are sent to the detectors with a delay represented by *T_cd0–5_* (Time Concentrator–Detector). This means that the *shutter* signals and *T0_Sync* signals (which reset the main ToA counter) are distributed with these delays. A sequential start of the detectors is used, introducing a 50 ns shift between consecutive detector layers.

While this approach contradicts strict time synchronization, it can be compensated for, and it provides benefits in terms of power supply stability and the designed approach for monitoring clock offsets in the nodes.

Once the individual detectors receive the shutter signal, which forces them to start the measurement, they resynchronize this shutter signal into the matrix clock domain, which is shared with ToA timing. Measurement then begins. The resynchronized shutter signal, which is now in the matrix clock domain, is subsequently sent via the PLL monitor output to the DC, where it is referred to as the *detector feedback* signal. The delays of these signals are denoted as *T_dc0–5_* (Time Detector–Concentrator) and consist of several important components:$$T_{{dc}_{n}}=T_{ss}+T_{mco}+T_{cable}+T_{pcb}$$
where*T_dcn_* is total delay between the Timepix3 shutter register and the DC, *n* representing the layer index;*T_ss_* stands for the sampling uncertainty of the shutter signal by the matrix clock (40 MHz);*T_cable_* represents the delay caused by the cable connecting Timepix3 and the DC (Ethernet cable), which is primarily determined by cable length and electrical parameters;*T_pcb_* includes the signal delay on the PCB as well as internal delays within the Timepix3. In total, it encompasses the delay from the shutter register in the chip to the connector mounted on the chipboard PCB.

Let us add that the absolute time at which the *detector feedback* signal arrives at the DC is also affected by the sequential start of the detectors. When the shutter is phased and delayed, it is clear that this timing will appear as an additional relative contribution to the delay of the detector feedback signal. However, since the shutter signal is resampled by the internal Timepix3 clock, this delay contribution may not be identical to the delay of the shutter signal caused by the postponed start of the detector.

The Pulse Generator in the DC receives detector feedback signals and produces a common time trigger signal, where the relative delays of the detectors are represented by square pulses. An example waveform of this signal (related to Section 5.2) is shown in Figure 6. Naturally, the Pulse Generator introduces an additional delay, denoted as *T_pd0–5_* (Time Delay of the Pulse Generator) in Figure 5, which must be considered. It is important to note that the Pulse Generator operates as an asynchronous logic circuit and does not rely on any clock signal or sampling.

Once *T_dc0–5_* and *T_pd0–5_* are characterized (authors use a signal generator with a fast oscilloscope or with a TDC device), they can be subtracted from the detector feedback arrival time for individual nodes:$$T_{{nodeoffset}_{n}}=T_{{fba}_{n}}-T_{{pd}_{n}}-T_{{dc}_{n}};$$
where*T_nodeoffset_* is final shift (compensation) of the ToA values in the Timepix3 node;*T_fba_* represents the time of detector feedback arrival at the DC;*T_pd_* is the delay introduced by the Pulse Generator;*T_dc_* is the total delay between the Timepix3 shutter register and the DC.*Note: n refers to the detector feedback channel.*

By applying this correction, we obtain time offsets for the ToA values in individual detector layers and compensate for delays caused by clock shifts and cabling. When absolute time is the subject of interest (e.g., in the case of an external trigger or when multiple measurement chains are used), additional delays, such as *T_dfb_* (Time Delay from Front-end to Back-end) and *T_dbtdc_* (Time Delay between the Back-end and TDC), also become significant. The demonstration in this paper focused on time compensation among detector layers only. Note that when multiple measurement chains are considered, a common clock distribution in the BEs of the chain is expected. Alternatively, another option is to use a more powerful BE to control multiple DCs and detector layers.

The device should also react to an external trigger. When an external trigger starts the measurement, the TDC measures the delays between this start event (typically the edge of the trigger signal) and the moment at which the detector layers start, as determined by the detector feedback signal.

Another typical case is the recording of timestamps for external events, such as a test beam trigger produced by a scintillator detector. For this purpose, we use an internal TDC implemented in the BE’s FPGA, which provides uniform timing with Timepix3 ToA values. Again, detector feedback signals are used for the initial alignment of the TDC counter measuring these external events.

If synchronization with a slower external clock is required—for example, the orbit clock of the LHC—the concept of measuring the rising edges of this clock as external events can be applied. The recorded timestamps (there is no need to record each timestamp of the rising edge) are then used for off-line compensation between the external clock and the local oscillator in the BE, which feeds the rest of the chain. This approach was used by the authors in the previous version (still operational) of the ATLAS TPX3 network, published in [6]. If synchronization with a fast external clock is needed, the PLL in the BE can be directly locked to this external clock.

## 5. Results

### 5.1. Laboratory Testing and Calibration

During the preparation for the setup, all six detector layers (silicion 500 µm) were calibrated. Threshold calibration, ToT-per-pixel calibration, and time-walk calibration were performed. A bias voltage of 200 V was applied to the sensor.

These processes involved irradiating the sensors with characteristic X-ray fluorescence photons from different elements (indium, zirconium, copper, nickel, and titanium) and a radioactive ^241^Am source. Subsequently, an energy versus ToT curve was constructed for each pixel, enabling the derivation of energy values from ToT measurements in individual pixels. Figure 7a shows a typical ^241^Am spectrum after calibration (for one of the detectors) and an example pixel calibration curve (Figure 7b). The dominant peak occurs at a mean value of 60.03 keV with a sigma of 1.64 keV (all clusters included). These values are typical for Timepix3 detectors with a 500 µm silicon sensor. Time-walk correction is crucial for maximizing the precision of ToA information since the signal skew of the amplifier output depends on the energy deposited in a pixel, which affects the timing of the signal crossing the threshold. The time-walk correction curve is shown in Figure 8.

### 5.2. Test Beam Measurement

The measurement chain was tested using a pion test beam with an energy of 180 GeV/c at the SPS facility at CERN. Three detector units (2× Timepix3) with silicon sensors, totaling six detector layers (hereafter abbreviated as Det0, Det1, …, Det5), were placed in a row behind each other so that their centers were irradiated by the particle beam. To verify the timing performance (via time-of-flight measurement), the distance between the first two units was 46 cm, and the distance from the second to the third was 5.68 m (see Figure 9). The DC and BE were interconnected by very short fibers; a Gigabit Ethernet interface was used for both control and data readout between the computer and BE.

During the measurement, several 30-min runs were taken. The detector configuration—including the matrix configuration, internal DAC registers, and control registers—was refreshed between runs. This reduces the probability of obtaining noisy data caused by SEUs in the control registers of the Timepix3 readout chip.

According to the position of the detectors relative to the beam line and the characteristics of minimum ionizing particles (MIPs), the majority of recorded clusters (particles) consist of a few pixels (2–5 pixels). However, not only primary beam particles hit the detectors, but also secondary particles, which appear as larger clusters—classified as heavy blobs and heavy tracks. A 200 ms frame of observed data from the beam spill is shown in Figure 10. All clusters and their energy values (ToT) are displayed on the left. Based on the values of cluster features (size, height, etc.), only MIP candidates that were filtered from the original measured data are shown in the right figure. The energy spectrum of these measured data is presented in Figure 11. During one beam spill (5 s), approximately 80,000 clusters were detected per detector, 98.8% of which are considered primary particles.

The main aim of the measurement and data evaluation was time-of-flight measurement to quantify clock distribution stability and the quality level of uniform timing in the system. Calibration recalculation and time-walk correction were applied to all measured data, after which the data were clustered. Before identifying particles hitting all six detector layers, we applied compensation for time delays and offsets among the nodes of the measurement chain, as described in Figure 9. An example of randomly selected particles going through all detectors is presented in Figure 12. Only three layers are depicted for better clarity. In Detector 2 and Detector 4, there are time-of-flight values from the measurements. The shown values are approximately within ±LSB of the Timepix3 time binning, which corresponds to the expected quantization error for this type of measurement.

From the reconstructed trajectories, it is evident that the alignment of the detectors with respect to the beam and to each other was not ideal (the detectors were not at the same height). This misalignment was caused by the imperfect installation of the detectors in the setup.

The obtained statistical results are summarized in Figure 13, which shows the time differences of clusters among the detector layers. The minimum pixel ToA value in the cluster represents the ToA of the entire cluster. It corresponds to the charge drift from the region closest to the pixel, meaning that the effects of sensor thickness are negligible.

These time differences should correspond to the time-of-flight time. Time differences among detectors increase with the distance between them. The first two detectors, which are positioned very close together (within a single housing), show a mean time difference of 0.54 ns. This is slightly above the theoretical timing resolution of Timepix3, which is 450 ps (calculated as 1.5625 ns/sqrt (12)).

On the other hand, significant delays were measured between Det 0 and Det 4 and 5.

Table 1 summarizes the results of the time-of-flight measurements. The table includes the distances between the detectors, the calculated and measured values of the time of flight (time differences of the measured clusters; mean and sigma of the Gaussian fit), and the errors between the measured and expected values. As previously mentioned, the delay between Det0 and Det1 should be close to zero, as these layers are practically in the same position. The second detector unit (Det2 and Det3) shows delays of 1.43 ns and 1.45 ns, respectively. These values closely correspond to the 46 cm distance between the detectors.

The distances between Det0 and Det4/Det5 are 6.14 m, corresponding to an expected delay of approximately 20.26 ns. The measured values align closely with this prediction: 20.59 ns for Det4 and 20.51 ns for Det5. Overall, all measurements exhibit errors smaller than one least significant bit (LSB) of the Timepix3 time counter and, except for Det0/Det1, remain within the theoretical Timepix3 timing resolution (450 ps).

All results presented in Table 1 were referenced to Det0 as the time reference. However, in total, fifteen detector pairs allow for time-of-flight (ToF) measurements, enabling the calculation of residuals. Table 2 summarizes these residuals as the difference between the calculated ToF value and the measured mean (obtained from a Gaussian fit). Notably, several values exceed the resolution of Timepix3 (450 ps; this will be discussed later). Based on these fifteen measurements, we determine a synchronization bias of 314 ps ± 56 ps.

As mentioned above, some results exceed the theoretical time resolution of Timepix3. This is also reflected in the standard deviations shown in Table 1 (ranging from 1.58 to 1.86 ns). It is important to note that additional sources of uncertainty contribute to the overall time resolution. The applied time-walk correction helps mitigate the time-walk effect; however, its residual RMS can still influence the results, as our time-walk analysis is performed at the chip level rather than for individual pixels. As shown in [23], this contribution can be approximately 450 ps.

Another factor negatively affecting the obtained results is the non-ideal behavior of the fToA counters. Previous studies [24] have demonstrated that fToA values differ significantly depending on whether the recorded hit is the first, second, or third within the super-pixel where the VCO is allocated for fast timing (640 MHz). The second and third hits can introduce delays up to six times longer than the first. Additionally, as reported in [25], a subset of pixels consistently fails to register hits across all fToA values.

Furthermore, Timepix3 suffers from clock distribution delays across the matrix. According to chip design specifications, this delay should be within 1.56 ns, introducing additional uncertainty if not properly compensated. Another well-documented issue is the delay variability within Timepix3’s internal building blocks, which fluctuates by up to 1 ns across groups of 16 rows [24,25]. Without these advanced compensation methods, the commonly achieved timing resolution is approximately 1.04–1.3 ns [23,24,25].

Comparing this with the results in Table 1, our resolution (recalculated from standard deviation of differences) falls within 1.12–1.31 ns, which is fully consistent with previously reported findings. In this discussion, we have not yet accounted for the influence of jitter in our distributed system. However, this contribution is expected to be minor, on the order of tens of picoseconds.

## 6. Deployment and Modifications

The described measuring chain with three detector units (six layers) has already been installed in several locations at CERN, including the ATLAS experiment cavern, the MoEDAL experiment at LHCb, and near the ALICE experiment at the LHC. The system has operated successfully, and the results of these measurements are planned for publication in 2025. The specific conditions of each installation required tailored modifications to the proposed systems. In other words, the practical utilization of the systems has provided valuable insights for further improvements.

In response to the unique magnetic field conditions in the ATLAS cavern (toroid area), a more advanced power supply concept for the DC was developed, as outlined in Chapter 4.2 and detailed in [26]. This power architecture is based on two pairs of radiation-hardened switching voltage regulators (see Figure 14)—one pair dedicated to providing the primary 3.3 V supply for FPGA operation and the other for powering the detector units. Each pair consists of a primary regulator and a backup unit. This redundancy ensures continued operation in the event of a failure in one of the converters. Additionally, both converters in a pair can be activated simultaneously to share the load, significantly enhancing the overall reliability of the power subsystem. For the FPGA’s additional power rails, LDO regulators are used. To supply power to the control and measurement circuits within the power unit itself, an onboard LDO regulator provides a stable 3.3V output. Given the requirement for reliable operation in strong magnetic fields, the switching power regulators utilize air-core inductors, high-speed switching, and an optimized controller. For this purpose, the CERN-developed ASIC bPOL48V_V2.1 was selected. Designed for operation with a GaN half-bridge power stage, this ASIC is specifically tailored for use in extreme environments [19,20]. In total, four such modules are implemented per DC power supply board. The design also meets fundamental diagnostic requirements, including current limits, over-voltage protection, and thermal protection.

During the deployment of the system at the LHC near IP2 (ALICE experiment), the use of metallic synchronization cables was not feasible. To address this, the authors made a slight modification to the measurement chain, specifically to the DC’s firmware. This adjustment eliminated the need for metallic cables between the DC and BE. Instead, the reference clock for the detectors is now derived from the data transmitted via fibers using Clock Data Recovery (CDR). A simple multi-phase clock TDC was implemented in the DC’s FPGA device, enabling the clock phase shifts among detectors to be measured directly within the DC. The final information about time offsets is then sent to the BE via the fibers. The removal of metallic cables also allowed for the use of significantly longer fibers between the DC and the back-end unit.

## 7. Conclusions

The presented system can operate with up to six Timepix3 detectors, offering a high data rate and precise time synchronization. As demonstrated in Chapter 5.2, the system supports time-of-flight measurements with an error of 314 ps ± 56 ps. Results suggest that the real time resolution of the Timepix3-based detector with silicon 500 um sensor is around 1.3 ns. Achieving such precision, however, requires careful attention to cabling, connectors, the time-walk correction of the detectors, uniform threshold settings, and other critical factors. However, authors are aware that results could be improved by the utilization of advanced compensation techniques (mainly compensation of time delays in columns).

Another significant advantage of the system—particularly useful for installations in large facilities—is the virtually unlimited distance between the front-end and back-end components of the readout. Its resistance to magnetic fields further enhances its applicability in environments with large magnets.

Finally, it is important to emphasize that this system is not a unique or final solution but rather a versatile platform that can be easily adapted to meet the specific requirements of different projects, environments, or facilities. For example, the synchronization concept described in Chapter 4.5 allows multiple Data Concentrators to be synchronized, enabling the extension of the detector network to include a larger number of detectors. Alternatively, the system can be simplified into a single-chip solution for scenarios requiring just one Timepix3 detector operating at a long distance from a safe area. Another possibility arises when a suitable FPGA device is used (the authors utilized the highly capable Intel Arria10), allowing for advanced on-line data processing directly within the back-end readout.

## Figures and Tables

**Figure 1 sensors-25-01860-f001:**
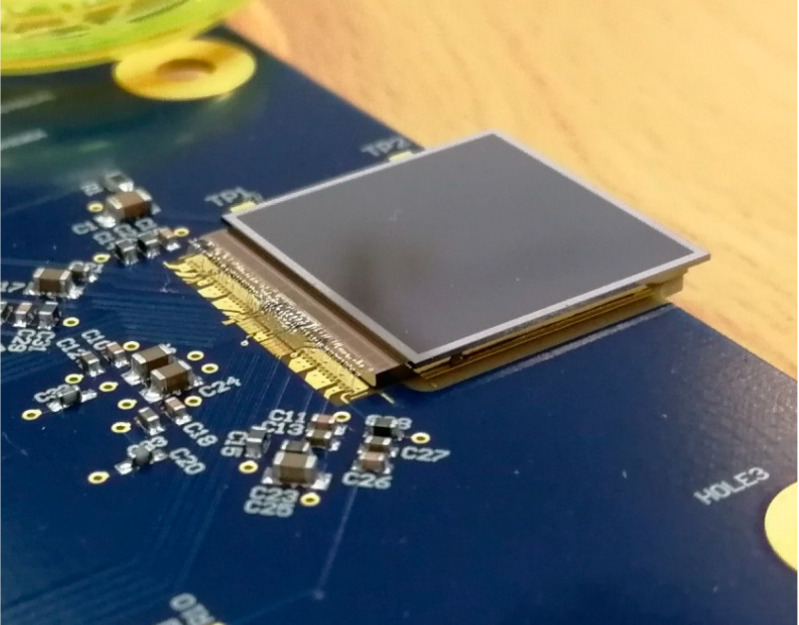
Timepix3-based detector glued on PCB. Wire bonds ensure interconnection with PCB.

**Figure 2 sensors-25-01860-f002:**
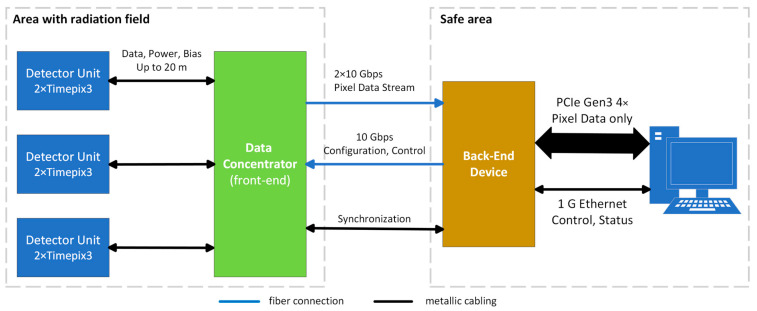
Proposed concept of measurement chain. The readout system is divided into two parts: the Data Concentrator (DC) with detector units, which are installed in the radiation field area, and the back-end device (BE) with a computer, located in a safe area free from radiation and magnetic fields. The connection between these parts is established via optical fibers (2 × 10 Gbps for pixel data and 1 × 10 Gbps for control and configuration).

**Figure 3 sensors-25-01860-f003:**
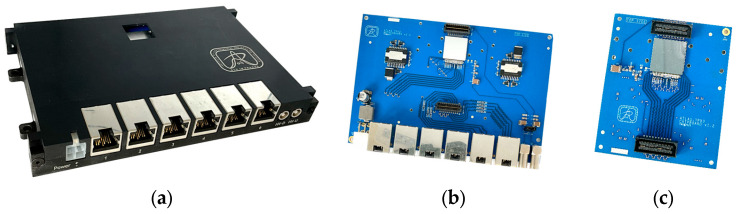
Detector unit. The unit consists of two boards: a mainboard (**b**) and a daughterboard (**c**). Each board hosts one Timepix3 detector and includes the required radiation-hardened power supplies. The unit is enclosed in an aluminum housing (**a**).

**Figure 4 sensors-25-01860-f004:**
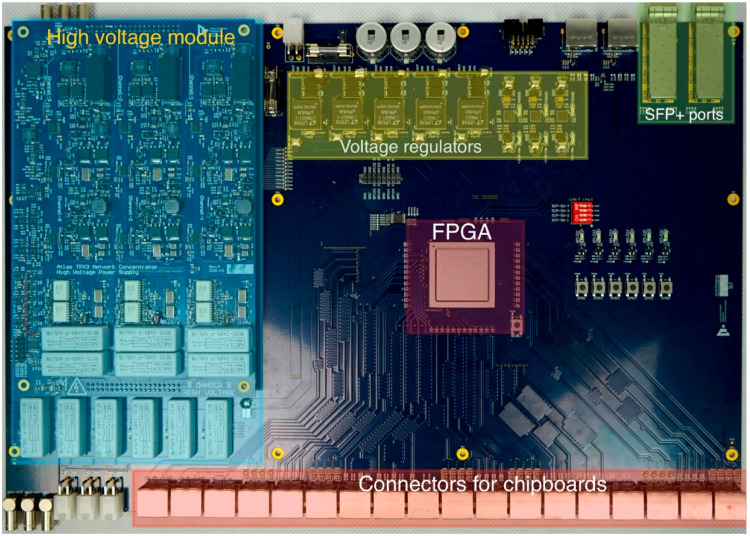
Data Concentrator. The relatively large PCB, designed to accommodate a high number of RJ-45 connectors, fits onto a 19-inch rack shelf for straightforward installation in a standard rack. The high-voltage module cannot be installed when a bypass and an external bias source are used.

**Figure 5 sensors-25-01860-f005:**
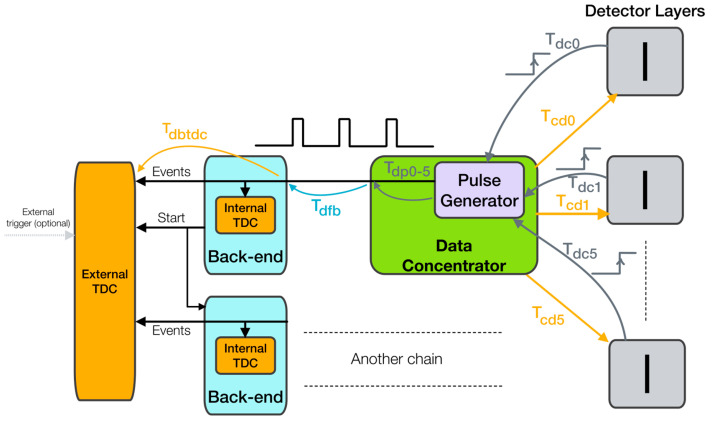
The synchronization scheme. The synchronization signal flow begins at the detector layers, where the detector feedback signal is distributed to the DC with a delay denoted as *T_dc0–5_*. The Pulse Generator then produces a common trigger signal that incorporates the relative delays among the detectors. This signal, further influenced by additional delays, is subsequently sent to the TDC, where it can be measured.

**Figure 6 sensors-25-01860-f006:**
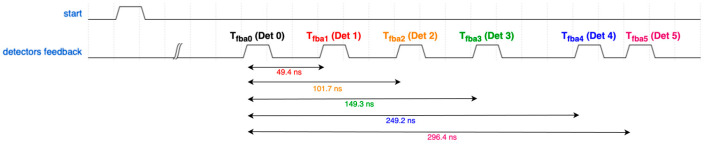
Output of the Pulse Generator (example from testing). The waveform illustrates time differences among pulses, representing the time delays between individual detector layers.

**Figure 7 sensors-25-01860-f007:**
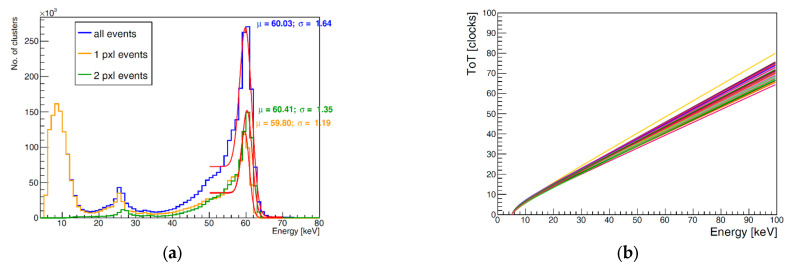
(**a**) Photon energy spectrum: response of the Timepix3 500 µm silicon detector to the 241 Am radiation source. The calibration and testing measurements were performed at a sensor temperature of approximately 60 °C. (**b**) Energy versus ToT curves for individual pixels. Each of the lines corresponds to a curve for one pixel.

**Figure 8 sensors-25-01860-f008:**
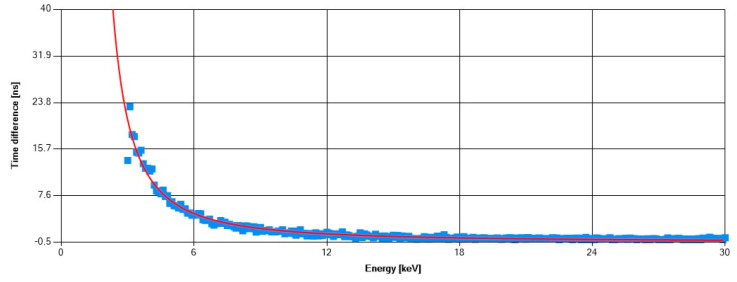
Time-walk correction curve. The graph shows that for energies below 12 keV, ToA correction is necessary. The correction value reaches up to 30 ns for pixel hits with very low energy.

**Figure 9 sensors-25-01860-f009:**
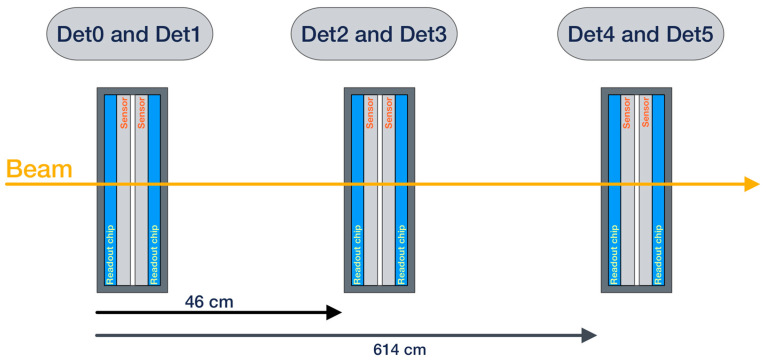
Arrangement of detectors in the experimental setup. The distance between the first pair of detectors is shorter (46 cm), while the distance to the third detector unit is significantly larger (614 cm).

**Figure 10 sensors-25-01860-f010:**
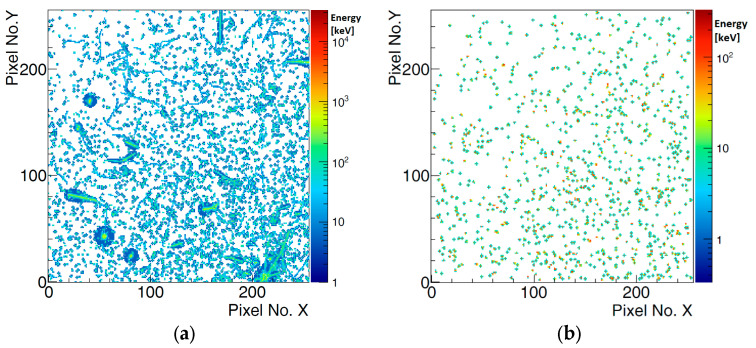
Recorded hits in Detector 0 (one layer). (**a**)—all clusters. (**b**)—only MIP candidates selected. The frames represent 200 ms of data during the beam spill.

**Figure 11 sensors-25-01860-f011:**
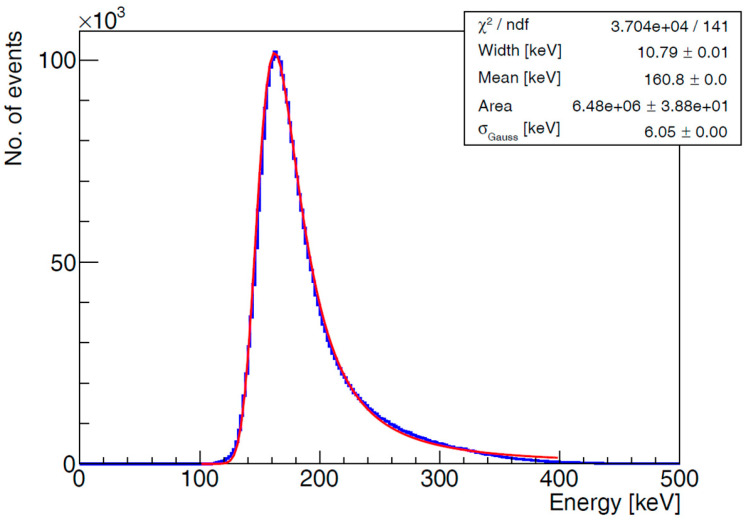
Energy spectrum of measured clusters (Detector 0). Calculated only for MIP candidates. Fitted with Landau and Gaussian functions.

**Figure 12 sensors-25-01860-f012:**
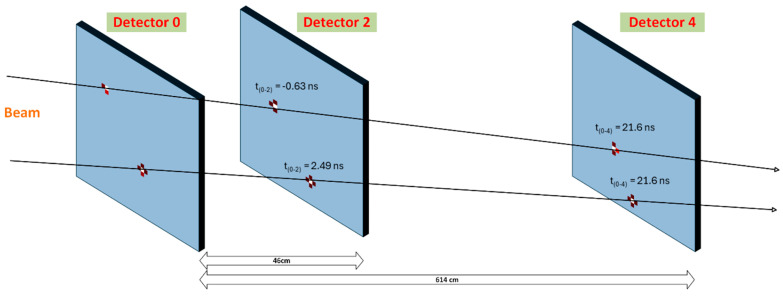
Two particles penetrate all six detectors in a row. For better clarity, only three detectors (the front detectors in the detector units) are shown. One can see clusters produced by particles in the detector layers. In Detector 2 and Detector 4, there are time-of-flight values from the measurements. From the reconstructed trajectories, it is evident that the alignment of the detectors with respect to the beam and to each other was not ideal (the detectors were not at the same height). Note: clusters (hit pixels) do not respect the dimensional scale related to the detector’s size (for better clarity).

**Figure 13 sensors-25-01860-f013:**
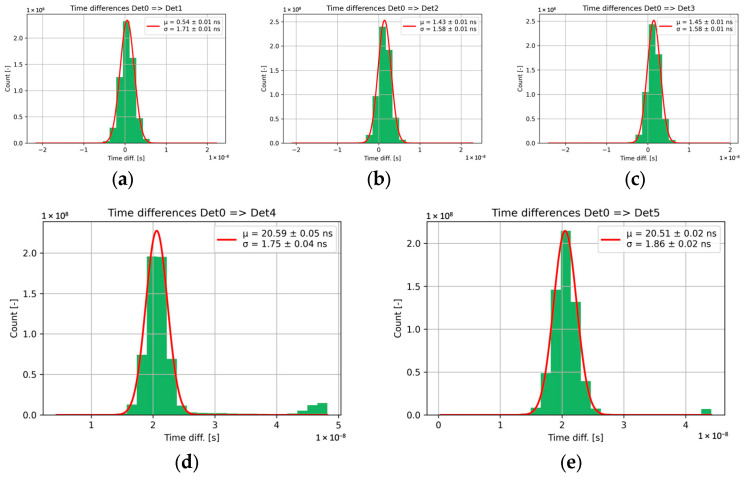
Time differences between clusters among detector layers. A histogram of time differences was constructed and fitted with a Gaussian function. (**a**) shows the time differences between Detector 0 and Detector 1. (**b**) shows the time differences between Detector 0 and Detector 2. (**c**) shows the time differences between Detector 0 and Detector 3. (**d**) shows the time differences between Detector 0 and Detector 4. (**e**) shows the time differences between Detector 0 and Detector 5.

**Figure 14 sensors-25-01860-f014:**
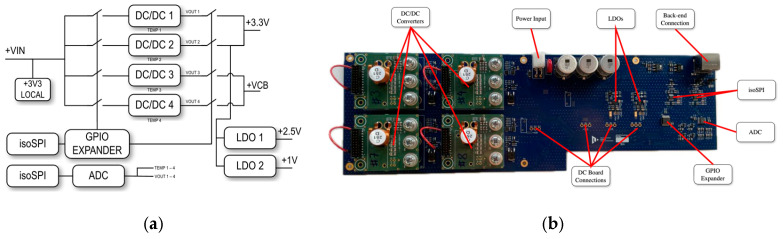
Power supply concept for system deployment in the ATLAS toroid area. (**a**) The power architecture is based on two pairs of radiation-hardened switching voltage regulators. (**b**) PCB implementation of the power supply management system. This board is integrated with the DC unit, replacing the original power supplies.

**Table 1 sensors-25-01860-t001:** Summary of time-of-flight measurement. The table shows all distances among detectors and results of measurement together with expected values.

Detectors	Det0 => De1	Det0 => De2	Det0 => De3	Det0 => De4	Det0 => De5
Distance [cm]	0	46	46	614	614
Time difference/Meas. ToF: mean [ns]	0.54	1.43	1.45	20.59	20.51
Time difference/Meas. ToF: sigma [ns]	1.71	1.58	1.58	1.75	1.86
Calculated Time of Flight [ns]	0.00	1.52	1.52	20.26	20.26
Error of measured ToF (Measured-Calculated) [ns]	0.54	−0.09	−0.07	0.33	0.25

**Table 2 sensors-25-01860-t002:** Time-of-flight measurements between all detector pairs. The table presents the residuals, defined as the differences between the expected and measured time-of-flight (ToF) values. All values are given in nanoseconds (ns).

Detector	D0	D1	D2	D3	D4	D5
D0	---	0.54	−0.09	−0.07	0.33	0.25
D1	---	---	−0.69	−0.44	0.02	−0.48
D2	---	---	---	−0.04	0.22	0.5
D3	---	---	---	---	0.73	−0.13
D4	---	---	---	---	---	0.18
D5	---	---	---	---	---	---

## Data Availability

Data are contained within the article.

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
