# Peer review of "Enhanced Readout System for Timepix3-Based Detectors in Large-Scale Scientific Facilities"

_sensors, 2025, doi:10.3390/s25061860_

Round 1
Reviewer 1 Report
Comments and Suggestions for Authors
Dear Authors,
congratulation for this nicely written paper and for successfully developing and testing this state of the art readout system for Timepix3 based readout.
General comments:
The paper is clear and the language is of good quality. The problematic is well explained, the approach to address it is well motivated and the technical description is clear and complete enough. The experimental setup and the calibration procedure is well explained and as well as the measurement. The results shows that the presented system can achieve very good performances. I have a few comments (2,5,6,8,9) that could help clarifying a few points to the reader.
The scientific methodology and approach is good (calibration/measurement method). My main scientific reservation concerns the estimation on the final synchronisation performance and the way it is quoted. Given the work the authors have put in conceptualising and building this system, the amount of work that was needed to do the experimental qualification (setup, calibration, measurement campain) and the seemingly good quality of the result, I believe a stronger quantification of the synchronisation performance would really benefit the paper, and this can achieve with rather minor work – see comment 7 - which is why I quoted “Can be improved” at the question “Are the conclusions supported by the results?”. I also put “Can be improved” in “Does the introduction provide sufficient background and include all relevant references?” for the comment #1 (should add reference to other similar readout system) and #8 (could add reference to detailed timing performance of Timepix3 based detector).
Comments and suggestions:
I have a few comments and questions and at the end I give a few wording suggestions.
-
In the introduction or in the proposed concept section, it would be good to refer to discuss other existing readout systems based on TPX3 and addressing similar problematics and the advantage of the one presented here. Not sure if there are many others but at least this one, used for beam gas imaging in the CERN PS, is relevant https://iopscience.iop.org/article/10.1088/1748-0221/14/01/C01013 (I don't know if this the most relevant publication, I could also find https://indico.cern.ch/event/965636/contributions/4063630/attachments/2141653/3608936/PS-BGI-TB-12-Nov-2020-JWS.pdf where the readout achitecture and components is shown slide 16).
-
In the "Proposed Concept" section it would be useful to quote what radiation level can be achieved: DU is likely limited by the Timepix3 itself, and the DC by the FPGA and the non-radiation hard tranciever, what is the dose at which they can be operated? Similarily do you have an estimation of the maximum magnetic field the system can operate with?
-
l91, could be intresting to highlight that seems not to be the case in the successors of the Timepix3 (both VeloPix and Timepix4 have higher datarate and can be/are used as optical transiver input).
-
l.171 is there a specific reason not to use the same DCDC than in the DU (eith the LHC4913 in DC or upgrade the DU to bPOL when upgrading the DC to bPOL)?
-
In "3.3 Back-end Device", are the LVDS signals used for synchronisation provided by the BE board? If not can you explain where they come from? Can it be used to synchronise with the accelerator clock (can you maybe add a few words about the possible sychronisation scheme)?
-
l.199 the synchronisation scheme is quite an important part of the system and I think the explanation could be expended a little. At least for me it is a bit unclear how all the part get together from the text (maybe a little diagram could also help understanding): the DC monitors the output of the PLLs of its DUs. What does it mean? Is the delay between the DUs PLL "readout" and corrected in the BE (or offline) or is the pulse generated for each DU and sent to the BE? If it is readout then is there just one pulse sent to the BE to "align" the DCs among themselves? Please go a little more in details to make this crucial section easier to understand. It would also be nice to see the typical amount of correction that has to be implemented between planes from the PLL monitoring and / or from the different cable length.
-
About the quantification of the synchronisation performance: The Timepix 3 lowest LSB account for 1.56ns ie. the best resolution that can be achieved by a Timepix 3 based detector is 450ps. One would expect a system designed to exploit the time measurements and relative time between Timepix 3 based detector to be (well) bellow the detector contribution.
In section 4 the study of the timing response is based on the time of flight measurement, and with 6 independent sensors there are 15 time difference to estimate a more robust average bias that can be achieved with the system. The paper would really benefit from having a more robust estimation of this critical parameter (and the data should be available already as suggested by Table 1).
Right now the reference to the result are a bit misleading: l301-302 "Overall, all measurements show errors smaller than one LSB of Timepix3's time resolution.". One LSB is not the Timepix3's time resolution, that would be 1 LSB/sqrt(12) ie. 450ps and at least one of the 5 measurement is larger. Again in the conclusion l. 332 you quote 1 of the 5 measurements (0.25ns) and say it is well below the LSB. Why this one, and again it is to be compared with 450ps which is the time resolution of TPX3.
Just to be clear the result seems really good, if we take the 5 "error of measured ToF" in Table 1 we obtain a bias from the synchronisation of 250ps+/-100ps and a standard deviation of 225+/-80ps (which would just add 10% to the best resolution achievable with Timepix 3) which is a nice result. This is not a proper statiscal approach because all measurements are correlated to Det0 but you have all the 15 time differences so you can get rid of the correlations and obtain a robust value which you can defend and which will make the result much stronger.
-
In section 4.1 you discuss the threshold and charge calibration as well as the timewalk correction. Did you considered also the various correction to the fTOA (per pixel, and per fTOA bins) which seems to be needed for precise timing measurement using Timepix3 (see for example https://iopscience.iop.org/article/10.1088/1748-0221/15/09/P09035/pdf, but there are also various presentations available on the web from Kevin Heijhoff). If not can you estimate the uncertainty, and more importantly if it could lead to a bias that could be contributing to your result? In this paper they also show that a time resolution of the order of 650ps can be achieved. From the relative time resolution you report in table 1 ("Time difference/Meas. ToF: sigma [ns]"), the mean is about 1.7ns (again not completely correct because all are correlated to Det0), so you'd expect a cluster time resolution of 1.2ns per plane. It would be useful to comment why the measurement is not at the same level as other timing measurement (I can imagine this is due to the tick planar sensor). I also believe you should refer to this paper (or another that investigate the timing performances of the Timepix3).
-
[related to the previous question] Can you specify the operation voltage of the sensor? With a 500um tick sensor the sensor timing resolution itself can be quite large. In https://iopscience.iop.org/article/10.1088/1748-0221/12/11/P11017 for 300um pad detector resolution is more than 0.5ns for HV below 100V. The resolution would be even worse for ticker sensors and for pixelated one so even with high stat it might put some uncertainty on your time of flight measurement.
Wording suggestions:
-
Overall the paper reads very well and the level of english is very good. Here are a few suggestions:
-
Abstract and over the document: maybe consistency between detector / sensor / based on TPX3 etc...
-
Title: [...] for Timepix3 based detectors [...]
-
l.10 readout system for Timepix3 based detector (Timepix3 itself is not a detector just a readout ASIC)
-
l.35 Timepix3 is the second generation of readout ASIC for pixel detector in the Timepix familly.
-
Section 2 title is "2. Timepix 3 Readout ASIC" and l35 to 57 indeed describe the Timepix 3 Readout ASIC, but then there is just 1 subsection "2.1 Long distance use" from line 60 to 85 describing the problematic about using the TPX3 in a long distance setup. Maybe there can be either another subsection for l.35 to 57 ("Main functionnalities" or something like that) or the section 2 could be named "Using the Timepix3 readout chip on long distance", without separting the asic description and the setting of the problematic.
-
l.71 "harm" --> "damage" (harm is more for living thing)
-
l.86 section 3 is numbered section 2
-
l.168 is it possible that "FE" or "DC" was meant instead of "BE"?
-
Fig 8, 9 and in the text: to avoid "only MIPs (probably)" in caption of fig8 experssion and the possible confusion (ex. in caption of fig9) between "real" MIPs and "probable" MIPs that come from your selection you could use the term "MIP candidate".

Author Response
In response to Reviewer 1:
1. Two references and a suitable comment were added to the Introduction.
2. A new paragraph was added to Chapters 4.1 and 4.2, including comments on expected doses.
3. I believe this comment referred to a different part of the text rather than line 191; however, I have added a note regarding this in Chapter 3.
4. Yes, the LHC4913 is a linear regulator. A switching power regulator is needed for DC applications. While the LHC4913 is partly obsolete, it has been used in this project due to previous positive experience and its strong utilization in the Medipix collaboration, as well as the availability of a significant stock. In the future, it is expected to be replaced.
5. Chapter 4.5 has been significantly extended.
6. Please see the extended Chapter 4.5.
7,8,9. Chapter 5 has been reworked.
Regarding wording suggestions, all recommendations were accepted.
Reviewer 2 Report
Comments and Suggestions for Authors
This paper reports a novel readout system for the Timepix3 detector, which overcomes challenges associated with long-distance operations in harsh environments, such as radiation and magnetic fields. The proposed design integrates front-end and back-end components linked via high-speed optical fibers. It offers high data rates, robust synchronization, and radiation resistance. The paper is well written, and its results are useful in the field of Timepix3 detectors. After addressing some minor revisions, It may publish in Sensors.
1. What do the author think are the potential limitations of the synchronization method for larger detector networks?
2. Please clarify the unique contribution of this work compared to reported works in literatures in the introduction.
3. Abbreviations such as "DC" without detailed definition are used in the paper, please check the whole paper and give the detailed definition
4. The influence of radiation on the long-term stability of the system needs more detailed exploration, particularly in terms of potential degradation of performance over time.
5. Please discuss more detailedly on the power management strategy in environments with significant magnetic fields.
6. The use of RJ-45 connectors in the system may be controversial due to their limited ability to handle high-radiation environments; more explanation is needed for their selection.
Author Response
In response to Reviewer 2:
1. Chapter 4.5 has been extended with a proper discussion on synchronization.
2. Additional text has been added to the Introduction.
3. The issue has been corrected in the text.
4. New text has been added to Chapter 4.1.
5. Chapter 6 has been extended, and the concept has been explained in more detail.
6. A note has been added to Chapter 4.1.
Round 2
Reviewer 1 Report
Comments and Suggestions for Authors
Dear Authors,
many thanks for this reviewed version, I am very happy with the answer you gave along the text to the few questions and comments. I also find the new additions very instructive.
I have a few comments on the new text, mainly language and formatting.
. 28 - 36. Not sure if it is intended as a comment for the review. Maybe it can be a bit better integrated to the text (here is a proposal, with modification highlighted in bold ):
“”"[…] radiation fields. While other similar systems exists [20],[21] they are designed for specific experimental conditions and rely on specialized components developed at CERN. The proposed system is intended to be more versatile, offering a choice between commercial and CERN-based components. Additionally, the authors aimed at designing a device compatible with their existing ecosystem, including chipboards and control software developed prevfiously [28]. Experience has shown that proper software support is crucial for the successful adoption of a device among users."""
l.104-106: thanks for the precision. Again here it is fine for an comment, but I would either remove it from the text or integrate it better:
“”"
[…] optical transceiver input. This will become possible with the latest version of the ASIC, the Timepix4 [add reference]."""
l.153-156: “While RJ-45 connectors massive electrical shielding may present challenges in environments with extremely high radiation levels , experience with these connectors in previous ATLAS
experiment setups [6][23] demonstrated they are reliable within the expected radiation conditions of this system.”
l. 163-164: I think Timepix3 designers are often uncomfortable about quoting that the ASIC has intriinsic radiation hardness. In the text you actually show that this is achieved mainly because you reconfigure teh ASIC frequently to “reset” the SEU effect. May I suggest to move the content of l.163-164 at the end of the paragraph (ie. l.174), so that it is clear it comes from those actions: “Based on our experience, and although some degradation is anticipated, the Timepix3 ASIC, sensor and associated chipboard electronics are able to withstand the level of radiation corresponding to the annual dose.”
section 4.5 sychronisation: Many thanks for adding all those very intresting details about the procedure. I think it is very useful infos.
l. 251: you introduce indexes 0 to 5. At that stage of the paper the reader have not yet seen the setup that you only introduce in section 5.2 (same for Fig 6). I am not quite sure what is the best way to adress that. Maybe line 250 you could say something like “The synchronization scheme of as system that would contain 6 detector planes, numbered 0 to 5, is shown in Figure 5.” or you could change the number to letter to stay generic, but will have to adapt the text in a few places.
l. 251, 252 there is a strange end of line.
l. 252 Maybe put Shutter and T0 Sync in texxtt or in italic to highlight they are signal names (the cap looks odd).
l. 253 The authors use a sequential --> A sequential start is used (it is fine to have this formulation in general but in the new text it is very present, maybe it can be modified in a few instance to make the text more fluid).
l. 266 […] domain (shared with ToA timing). --> which is shared with ToA timing
l.267 (now in the matrix clock domain) --> which is now in the matrix clock domain
l 292 Chapter --> Section
l 299 to 308, some of the subscripts are in italic, some not
l 322-325 proposal: “The device should also react to external trigger. When an external trigger starts the measurement, the TDC […] and the moment at which the detector layers start, as detemrined by the detector feedback signal”
l. 427: 1.5625ns/sqrt(12)
Congratulation again for this nice work and the obtained results!
Author Response
Dear Reviewer,
My answers above (almost fully accepted):
. 28 - 36.
Text slightly modified.
l.104-106:
Subject of the comment was removed.
l.153-156:
Rephrased.
- 163-164:
Text moved. Slight changes to text.
- 251:
Added explanatory text referring to section 5.2. Error in figure (layers numbered 1-6) found and corrected.
- 251, 252:
Fixed.
252:
Highlighted.
Suggestions above fully accepted and fixed in the revision:
- 253 The authors use a sequential --> A sequential start is used (it is fine to have this formulation in general but in the new text it is very present, maybe it can be modified in a few instance to make the text more fluid).
- 266 […] domain (shared with ToA timing). --> which is shared with ToA timing
l.267 (now in the matrix clock domain) --> which is now in the matrix clock domain
l 292 Chapter --> Section
l 299 to 308, some of the subscripts are in italic, some not
l 322-325 proposal: “The device should also react to external trigger. When an external trigger starts the measurement, the TDC […] and the moment at which the detector layers start, as detemrined by the detector feedback signal”
- 427: 1.5625ns/sqrt(12)
Best regards,
Petr.